# Small Molecule RBI2 Disrupts Ribosome Biogenesis through Pre-rRNA Depletion

**DOI:** 10.3390/cancers15133303

**Published:** 2023-06-23

**Authors:** Catherine E. Scull, Guy Twa, Yinfeng Zhang, Naiheng J. Yang, Robert N. Hunter, Corinne E. Augelli-Szafran, David A. Schneider

**Affiliations:** 1Department of Biochemistry and Molecular Genetics, University of Alabama at Birmingham, Birmingham, AL 35294, USA; 2Southern Research, Birmingham, AL 35205, USA

**Keywords:** ribosome synthesis, RNA polymerase, transcription, RNA decay

## Abstract

**Simple Summary:**

Ribosome biosynthesis has emerged as an excellent target for inhibition of cancer cell growth and proliferation. We used a high throughput screen to identify potent inhibitors of ribosome synthesis in malignant melanoma cells. Here, we show that the newly identified ribosome biogenesis inhibitor, RBI2, inhibits multiple cancer cell types by a mechanism of action that is distinct from previously described compounds. Using chromatin IP, isotopic labelling, and RNA sequencing, we conclude that treatment with RBI2 does not inhibit transcription initiation by RNA polymerase I, but appears to induce rapid polyadenylation and degradation of ribosomal RNA. This mechanism of action would be unique from previously described inhibitors of RNA polymerase I.

**Abstract:**

Cancer cells are especially sensitive to perturbations in ribosome biogenesis as they rely on finely tuned protein homeostasis to facilitate their rapid growth and proliferation. While ribosome synthesis and cancer have a well-established relationship, ribosome biogenesis has only recently drawn interest as a cancer therapeutic target. In this study, we exploited the relationship between ribosome biogenesis and cancer cell proliferation by using a potent ribosome biogenesis inhibitor, RBI2 (Ribosome Biogenesis Inhibitor 2), to perturb cancer cell growth and viability. We demonstrate herein that RBI2 significantly decreases cell viability in malignant melanoma cells and breast cancer cell lines. Treatment with RBI2 dramatically and rapidly decreased ribosomal RNA (rRNA) synthesis, without affecting the occupancy of RNA polymerase I (Pol I) on the ribosomal DNA template. Next-generation RNA sequencing (RNA-seq) revealed that RBI2 and previously described ribosome biogenesis inhibitor CX-5461 induce distinct changes in the transcriptome. An investigation of the content of the pre-rRNAs through RT-qPCR revealed an increase in the polyadenylation of cellular rRNA after treatment with RBI2, constituting a known pathway by which rRNA degradation occurs. Northern blotting revealed that RBI2 does not appear to impair or alter rRNA processing. Collectively, these data suggest that RBI2 inhibits rRNA synthesis differently from other previously described ribosome biogenesis inhibitors, potentially acting through a novel pathway that upregulates the turnover of premature rRNAs.

## 1. Introduction

Interest in nucleolar variations in tumor cells has persisted for over a century [1]. In the early studies, one of the first identified features of cancer cells was an increase in size and/or number of nucleoli [1,2]. It was hypothesized that this observed increase in nucleolar size/number in cancer cells is elicited by the characteristic increase in demand for protein synthesis observed in the majority of cancer cells [1,3,4]. Ribosome biogenesis, the primary process that takes place in the nucleolus, is a highly complex, energy-intensive, multi-step pathway [5]. During phases of rapid growth, ribosome biogenesis is the most energetically costly process in the cell, demanding tight regulation of the process [6]. This tight regulation involves the monitoring of free ribosomal binding proteins and/or free rRNA. In instances where pre-rRNAs are dramatically limited, these imbalances can ultimately result in cell death via p53-mediated apoptosis [2].

The exploitation of cancer cells’ dependence on ribosome synthesis has emerged as an attractive target for cancer treatment in recent years [6,7,8,9]. There are currently small-molecule compounds in clinical (CX-5461) and pre-clinical (BMH-21) trials that target cancer cells’ sensitivity to ribosome biogenesis [7,10,11,12,13,14]. There have been additional alternative molecular targets characterized for CX-5461, possibly due to distinct cellular contexts [15], but it is notable that both CX-5461 and BHM-21 appear to inhibit the binding of Pol I to the rDNA. CX-5461 prevents Pol I transcription initiation, whereas BMH-21 acts via DNA intercalation into the rDNA and subsequent signaling for the degradation of Pol I’s largest subunit [7,12,14].

In a recent study, we identified a new class of ribosome biogenesis inhibitors (ribosome biogenesis inhibitors 1 and 2, or RBI1 and RBI2) through a series of high-throughput screens [16]. We further demonstrated that RBI2 inhibits colony formation in soft agar experiments and that RBI2 specifically inhibits the growth of cancer cells, but has no effect a non-cancerous, non-transformed control cell line (HUVEC cells). At least two pressing questions remain regarding the mechanism of cancer cell growth inhibition via RBI2: (1) how does RBI2 down-regulate ribosome biogenesis, and (2) is RBI2’s mechanism of action distinct from the previously described mechanisms of the inhibitors CX-5461 and BMH-21?

In this study, we demonstrate that RBI2 potently inhibits cancer cell viability and rRNA synthesis, while also demonstrating that it does not affect Pol I occupancy of the ribosomal DNA (rDNA). To gain insight into the global effects of RBI2 on mRNA expression, we conducted RNA-seq analyses in MDA-MB-231 cells after treatment with CX-5461 or RBI2 and analyzed the results as a function of time. RNA-seq analysis revealed that CX-5461 and RBI2 alter gene expression distinctly in MDA-MB-231 cells. Further, we observed an increase in polyadenylated rRNA after treatment with RBI2, which was not observed for the cells treated by CX-5461. Finally, we noted that RBI2 does not impact normal pre-rRNA processing pathways. Though the specific molecular target of RBI2 requires further investigation, it is clear that RBI2 induces the polyadenylation of rRNA and thus likely inhibits cancer cell growth via a mechanism distinct from that of CX-5461. The distinct nature of RBI2 activity suggests the compound may provide a novel avenue for ribosome biogenesis inhibition and thus merits further development as an inhibitor of cancer cell growth.

## 2. Materials and Methods

### 2.1. Standard Growth Curves and Cell Viability

Alamar blue standard curves and cell viability assays were created and performed, respectively, for the MDA-MB-231 and HCC1937 cell lines (purchased from ATCC) exactly as previously described for A375 cell lines [16]. Briefly, 5000 cells/well were resuspended in 1X DMEM with 10% FBS media (Thermo Fisher, Waltham, MA, USA) to a total volume of 90 µL in a 96-well plate and incubated overnight at 37 °C with 5% CO_2_. The following day, 10 µL of CX-5461, RBI2, or RBIX was added per well to a final concentration ranging from 10 nM to 10,000 nM. Cells were incubated at 37 °C with 5% CO_2_ for 72 h. Cell viability was measured using the Alamar Blue assay (Thermo Fisher, Waltham, MA, USA).

### 2.2. RT-qPCR

Two hundred thousand cells were treated with 5 µM of RBI2 or CX-5461 in a 6-well plate (Falcon cat# 353046) and analyzed as a function of time. CX-5461 is an established Pol I transcription inhibitor; thus, it served as a positive control. DMSO (vector) was included as a negative control. RNA was purified via TRIzol extraction (Life Technologies #10296028) followed by Purelink RNA mini Kit (Invitrogen cat# 12183018A). Purified RNA was reverse-transcribed using Superscript Reverse Transcription kit (Invitrogen cat# 18080093) and amplified with either random hexamers or Poly-dT oligonucleotide. For pre-rRNA level quantifications, 5′ETS was amplified to produce cDNA, which was quantified and normalized to GAPDH levels. Primer sequences are presented in Table 1.

### 2.3. Chromatin Immunoprecipitation (ChIP)

One million exponentially growing cells were inoculated into one 10 cm diameter petri dish. Cells were incubated at 37 °C with 5% CO_2_ overnight. Cells were treated with 1 µM or 5 µM of RBI2 or CX-5461 for 30 min or 2 h. Then, formaldehyde was directly added to the media to reach a final concentration of 1% and incubated at room temperature for 10 min. Glycine was added to a final concentration of 0.125 M, and cells were incubated at room temperature for 5 min to quench the cross-linking reaction. Media were removed from plates and cells were washed with equal volume of cold 1X PBS. A total of 5–8 mL of cold Farnham lysis buffer (5 mM PIPES pH 8.0, 85 mM KCl, 0.5% NP-40, and freshly added protease inhibitor) was added per plate. Cells were scraped and transferred into 15 mL conical tubes placed on ice. Crude nuclei were pelleted at 4 °C, snap-frozen in liquid nitrogen, and stored at −80 °C until use. The following steps pertaining to ChIP were performed as previously published [17], except that chromatin was sheared at 4 °C using a Bioruptor (Diagenode, Denville, NJ, USA) for 30 cycles of 30 s on and 30 s off per cycle. A total of 2 µg of anti-A194 antibody (Santa Cruz cat# 48385) was used to immunoprecipitate Pol I complexes from about 500 µg of total protein. Quantitative PCR was used to determine Pol I occupancy of the rDNA at the promoter, 5′ETS, 18S, 28S, and IGS. GAPDH locus was used as a negative control. Primer sequences are presented in Table 2. Significance was calculated using a t-test with an FDR (Q) of 1%.

### 2.4. RNA-Seq Assays

One million cells were incubated overnight and then treated with 5 µM of RBI2 or CX-5461 and analyzed as a function of time (10 min, 30 min, and 120 min). Two replicates were included for each time point. Another two samples were not treated and served as untreated controls. RNA was extracted with TRIzol followed by Purelink RNA mini kit. Integrity of RNA samples was assessed using an Agilent Bioanalyzer 2100. RNA-seq libraries were prepared using KAPA Stranded mRNA-Seq kit. Briefly, mature mRNA was isolated from total RNA, fragmented, and then reverse-transcribed into cDNA. Illumina Tru-seq adapters were ligated to the ends of the cDNA. The library was amplified, and 25 million reads per sample were sequenced using the Illumina Sequencing Platforms HiSeq2000 at the University of Alabama at Birmingham, AL.

To validate the compound’s effects on pre-rRNA observed through RT-qPCR, rRNA RNA-seq reads were first aligned to a complete human ribosomal RNA repeating unit (GenBank: U13369.1). Coverage of the ribosomal DNA and the 5′ ETS in each sample were compared in IGV (PMID: 21221095 [18]). This allowed for isolation of “contaminating” rDNA reads corresponding to the polyadenylated pre-RNA observed via RT-qPCR. 

Following alignment of ribosomal RNA reads, the remaining unaligned reads were mapped to GRCh38 transcripts using Salmon (PMID: 28263959 [19]). Gene-level summaries were prepared from these transcript quantifications using tximeta (PMID: 32097405 [20]) and imported into DeSeq2 for differential expression analysis. Experimental Samples were compared at each time point to the control samples. Volcano plots were generated using EnhancedVolcano [21]. Genes with a p-value below 0.05 and an LFC ≥ 1 or ≤−1 were considered significant and retained for use in Venn diagrams and volcano plots. Overrepresentation analysis (ORA) of biological process GO terms was then conducted using WebGestalt (source: https://doi.org/10.1093/nar/gkz401 (accessed on 6 February 2022)) with the default parameters, with differentially expressed gene lists and the human genome used as a reference set.

### 2.5. RNA Gel Electrophoresis, Visualization, and Northern Blotting

Two hundred thousand MDA-MB-231, A375, or HCC1937 cells in a total volume of 2 mL were inoculated into each well of a 6-well plate in phosphate-free RPMI-1640 medium and allowed to attach overnight. Cells were then harvested, and RNA was purified with TRIzol in combination with Purelink RNA mini kit. Isolated RNA was then run on a 1% formaldehyde/MOPS:agarose gel and transferred to a membrane for probing, as previously described [22]. Briefly, total RNA concentration for each time point was normalized to ~5 µg/lane and, subsequently, mixed in a 1:1 ratio with loading dye. The loading dye/RNA mix was heated to 95 °C for 5 min and then placed directly on ice for at least 5 min. The RNA was electrophoresed on a 1% formaldehyde/MOPS:agarose gel. Gels were imaged, washed four times for 5 min with deionized water, and then incubated in MOPS 1X buffer for 20 min. Subsequently, RNA was transferred from the gel to a GeneScreen membrane. Membranes were then probed with radiolabeled oligonucleotide P3 [23]. The primer sequence used for Northern blotting is listed in Table 3.

## 3. Results

### 3.1. RBI2 Inhibits Cancer Cell Growth Similarly to Positive Control CX-5461

Several labs have demonstrated that ribosome biogenesis inhibition effectively inhibits the growth of numerous cancer cell types [6,7,8,9,10,24,25,26,27]. Nascent ribosomal RNA (rRNA) synthesis using Pol I is an attractive target for ribosome biogenesis inhibition as it is the first and rate-limiting step in ribosome biogenesis. Previously, our lab identified a small-molecule ribosome biogenesis inhibitor, RBI2, through a series of high-throughput screens [16]. To examine whether RBI2 inhibits breast cancer cell growth and proliferation similarly to the previously published RBI2-mediated inhibition of A375 cell growth, Alamar blue cell viability assays were performed for both the MDA-MB-231 (Figure 1) and HCC1937 cell lines (Appendix A). Cells were treated with a series of concentrations of RBI2, the positive control CX-5461, and the negative control RBIX. The positive control, CX-5461, is a known inhibitor of rRNA synthesis that putatively functions by inhibiting the transcription initiation of Pol I [7,10]. The negative control, RBIX, is structurally similar to RBI2 but lacks inhibitory activity [16]. Indeed, the inhibition of MDA-MB-231 and HCC1937 cell growth was similar to that observed for A375 cells after treatment with RBI2 [16], demonstrating that RBI2 potently inhibits growth of multiple types of cancer cells in vitro. These observations contrasted with the findings from the Alamar blue experiments in the presence of RBIX, which did not alter cell viability (Figure 1).

### 3.2. Ribosome Biogenesis Inhibitor RBI2 Rapidly Decreases Pre-rRNA Levels in Cancer Cells

We previously demonstrated that RBI2 inhibited cell viability and decreased pre-rRNA levels in A375 cells, which are malignant melanoma cells [16]. In this study, we established time courses for the RBI2 treatment of MDA-MB-231 and HCC1937 cells and validated the RBI2 inhibition of pre-rRNA levels in breast cancer cell lines. A375 cells were included as a positive control. In order to measure changes in pre-mature rRNA over time, RT-qPCR measurements of the 5′ externally transcribed sequence (ETS) levels were taken for the A375, MDA-MB-231, and HCC1937 cells treated with RBI2 over a series of time points (Figure 2A–C). The 5′ETS region of the rRNA was used because it is rapidly processed and is thus a good indicator of pre-rRNA levels. We found that in all cell lines, treatment with RBI2 induced a rapid reduction in the abundance of pre-rRNA levels over time (Figure 2).

To further validate our RT-qPCR results, RNA isolated from cells was run on denaturing agarose gels and, subsequently, stained with ethidium bromide (Appendix A). There was a clear loss of the 45S/47S pre-rRNA signal over the length of the time-course, which is consistent with the decrease in 5′ETS observed in the RT-qPCR experiments. In contrast to the 45S/47S pre-rRNA, the 28S and 18S rRNAs are mature rRNAs and thus function as an indicator of the steady-state levels of cellular rRNAs. While we observed rapid inhibition of pre-rRNA abundance in all three cell types, there were no observed changes in the abundance of mature rRNAs. These data confirm that RBI2 effectively and rapidly decreases pre-rRNA levels compared to GAPDH levels over time and does so without inducing the turnover of mature rRNAs (Figure 2A–C). We noted a significant reduction in pre-rRNA after 20 min of treatment and the complete loss of the pre-rRNA signal after 60 min of treatment. The observed rapid decrease in the pre-rRNA signal supports the hypothesis that the effect of RBI2 on ribosome biogenesis may be direct.

### 3.3. RBI2 Does Not Alter Pol I Occupancy of the rDNA

After noting that pre-rRNA levels were altered, we were curious as to the stage at which the pre-rRNAs were lost. There are several possibilities: transcription initiation could be inhibited, the transcription elongation rate could be reduced, or pre-rRNA decay could be induced. Chromatin immunoprecipitation (ChIP) experiments were performed to test if RBI2 impacted Pol I transcription by probing the occupancy of the polymerase on the rDNA (Figure 3). In these experiments, cells were grown until reaching 75% confluence and then treated for 120 min with RBI2 (the time point at which rRNA levels were completely depleted in the RT-qPCR experiments). After 120 min of treatment, there were no substantial changes in polymerase occupancy observed in the rDNA in the MDA-MB-231, HCC1937, or A375 cell lines (Figure 3A–C, respectively), suggesting that either RBI2 does not directly influence Pol I transcription initiation on the rDNA or, if it does, it impacts the rates of all stages of Pol I transcription (initiation, elongation, and termination) equally.

We then carried out Pol I ChIP across a series of time points (0, 10, 40, 60, and 120 min) to ensure that there was no transient alteration in Pol I occupancy over time, which could have been missed in our initial ChIP experiments. We again saw no significant alterations in occupancy at the promoter or at the 18S region of the rDNA at any time point (Appendix A). These observations are strikingly different from those of previous studies incorporating the ribosome biogenesis inhibitors CX-5461 and BMH-21, which both dramatically reduce Pol I occupancy on the rDNA [7,11]. These data suggest that it is unlikely that RBI2 alters Pol I affinity to rDNA or the initiation of transcription and further demonstrate that RBI2 inhibits ribosome biogenesis via a mechanism distinct from CX-5461 and BMH-21.

### 3.4. RNA-Seq Analysis

The experimental approaches described above and previously [16] have elucidated the general mechanism by which RBI2 inhibits ribosome biogenesis. These data strongly indicate that RBI2 perturbs ribosome biogenesis in cells by decreasing pre-mature rRNA levels. However, it remained unclear whether rRNA was the sole target of RBI2 or if other genes’ expressions were also altered by RBI2. To identify the effects of RBI2 on global gene expression, RNA-seq was used to compare differential gene expression pre- and post-treatment with either RBI2 or the positive control compound CX-5461. Out of 52,709 human genes, fewer than 1% of genes were affected by RBI2 or CX-5461 treatment after 10 min of treatment (Figure 4). These data suggest that neither RBI2 nor CX-5461 acutely impair RNA polymerase II activity or mRNA turnover.

Volcano plots were generated to visualize the gene expression results (Figure 4A). The volcano plots demonstrate that the MDA-MB-231 cellular transcriptome is altered within 10 min of exposure to both RBI2 and CX-5461. While RBI2 appears to induce a very slight increase and subsequent decrease (10 min: 418 genes, 30 min: 432 genes, and 120 min: 284 genes) in the number of genes with altered expression, CX-5461 appears to generate a slight decrease and a subsequent increase (10 min: 280 genes, 30 min: 221 genes, and 120 min: 498 genes) in the number of genes with altered expression over time (Figure 4A,B). We examined genes that were differentially expressed across all time points following the RBI2 treatment and found 84 overlapping genes (Appendix A). No obvious trend was evident in the differentially expressed genes between the two treatments, but the kinetics of the observed feedback responses to the treatment were notably different. The transcripts that demonstrated significantly altered abundance had a relatively small overlap between CX-5461 and the RBI2 treatment (Figure 4B), supporting the idea that these two compounds have unique molecular targets. Principle component analysis further highlighted the differences between the CX-5461 and RBI2 treatments, where the largest difference observed occurred between CX-5461 and RBI2 replicates after 30 min of treatment (Figure 4C).

We initially attempted to assess cellular responses to the RBI2 and CX-5461 treatments at each time point with respect to our DMSO control treatment; however, gene ontology assessment did not reveal significant changes in GO terms with respect to the control. Additionally, we did not find significantly enriched GO terms for the 84 genes differentially expressed across the time points following RBI2 treatment. Thus, to further explore the different cellular responses these two compounds illicit, we conducted differential expression analysis comparisons for the samples between treatment groups at each time point (Appendix A). At each time point, the replicates were clearly segregated by treatment type, with the greatest variance observed at thirty minutes post-treatment. The top 10 significant (FDR < 0.05) enriched GO terms for each time point and treatment group were reported (Appendix A). These enriched gene sets highlight RBI2’s nuanced activity, as GO terms associated with the negative regulation of RNA transcription, metabolism, and biosynthetic processes were consistently enriched over time for RBI2-treated cells after 10, 30, and 120 min of treatment. In contrast, the CX-5461-treated cells demonstrated an increase in enrichment in categories related to metabolic and biosynthetic processes after 10 min of treatment, cell stress and cell cycle regulation at 30 min of treatment, and RNA biosynthesis and transcription-related responses occurring after 120 min of treatment with CX-5461.

### 3.5. RBI2 Treatment Results in an Increase in the Poly-Adenylation of Pre-Mature rRNAs

Interestingly, RNA-seq alignment quality control analysis indicated that the RBI2-treated samples contained substantially more rRNA than the non-treated samples (Figure 5A). This was a surprising finding, as mRNA enrichment was executed using Poly-dT beads, which should result in rRNA depletion since rRNAs are not typically polyadenylated. Thus, as seen in the DMSO-treated negative control and the CX-5461-treated positive controls, the amount of rRNA recovered post-purification should have been minimal and consistent between treatments. Nonetheless, we observed reproducible rRNA recovery in the RBI2-treated cells at all time points (Figure 5A). Based on these observations, we hypothesized that RBI2 treatment might result in pre-rRNA polyadenylation.

To ensure that the observed increase in rRNA in our RNA-seq library post-polyA pulldown was not simply due to contamination from rRNAs during mRNA enrichment, the RBI2-treated samples were tested further for polyadenylation. RT-qPCR was performed on rRNA collected from the A375 and MDA-MB-231 cell lines using a Poly-dT oligo for reverse transcription instead of random hexamer primers. This strategy resulted in the reverse transcription of rRNAs only in cases wherein the rRNA was poly-adenylated. qPCR was then executed using oligonucleotides specific for the 5′ETS region of the rRNA (probing pre-rRNA levels). In both the A375 and MDA-MB-231 cells, a rapid increase in the polyadenylation of pre-rRNA at the 10 min time point was observed, followed by the decay of the pre-rRNA signal (which is consistent with the observed decrease in pre-rRNA; Figure 2 and Appendix A).

Deeper probing of the sequence alignments for poly-A containing rRNAs revealed that robust polyadenylation occurs across the rDNA at all treatment time points for RBI2, especially when compared to untreated and CX-5461-treated cells (Figure 5B,C). In the context of previous published literature, these data support a model in which depletion of the pre-rRNA is mediated by poly-A signaling dependent pre-rRNA turnover [2]. This turnover could potentially be accomplished through the TRAMP-nuclear exosome pathway, which has been demonstrated to degrade poly-adenylated pre-rRNA in response to transcriptional stress [22,28]. These observations lead to the hypothesis that the mechanism by which RBI2 inhibits pre-rRNA levels is through polyadenylation and the subsequent TRAMP–exosome-mediated degradation of the pre-rRNA.

### 3.6. RBI2 Treatment Does Not Result in Altered rRNA-Processing Pathways

As there was no observed impact on Pol I occupancy or clear impairment of RNA synthesis upstream of RNA processing, we questioned whether RBI2 instead inhibits or alters the downstream processing of pre-rRNAs. Previous reports have suggested that the altered processing of rRNA can result in TRAMP-mediated degradation of rRNA and decreased cell growth [22,29]. To determine whether RBI2 treatment influenced rRNA processing events, we executed Northern blotting analysis of the pre-rRNA isolated from the A375, MDA-MB-231, and HCC 1937 cells treated with RBI2 over different time periods (Figure 6 and Appendix A). If one or more steps in pre-rRNA processing is impaired, one would expect the reassortment of rRNA precursor abundance in response to the RBI2 treatment. Although a significant decrease in pre-rRNA abundance was observed over time (as expected according to our RT-qPCR experiments shown in Figure 2), a gradual decline in all rRNA precursors was observed, suggesting that it is unlikely that RBI2 generates “roadblocks” to downstream processing events. Furthermore, we did not observe switching of the cleavage sites to any of the known alternative cleavage pathways that have been demonstrated to result in the downregulation of ribosome biogenesis [2,22,29,30]. These data suggest that RBI2 does not impact the rRNA-processing pathway but instead induces the polyadenylation-mediated degradation of pre-rRNAs through an alternative mechanism. In the future, more exhaustive characterizations of the acute effects of RBI2 are required to isolate the exact molecular target with which the compound specifically decreases rRNA abundance.

## 4. Discussion

### 4.1. RBI2 Mediates Ribosome Biogenesis Inhibition by Decreasing Pre-rRNA Levels

The data presented herein clearly demonstrate that RBI2 treatment results in a decrease in pre-rRNA levels in mammalian cell cultures. As previously described, RNA turnover is required to maintain cellular homeostasis in response to environmental stress [2,22,29]. In mammalian cells, the half-life of mature rRNA is as long as several days [31,32]; however, in this study, we observed that RBI2 treatments as short as 10 min resulted in a significant increase in polyadenylated rRNA and a subsequent decrease in overall pre-rRNA (Figure 2 and Figure 5). These findings suggest that RBI2 abruptly triggers the turnover of newly synthesized rRNA, which likely occurs through the polyadenylation of the pre-rRNA. Overall, this rRNA depletion leads to the loss of cell viability in cancer cells but does not significantly impact the growth and viability of non-cancerous control cells [16]. Presumably, normal cells retain the regulatory capacity necessary to overcome the induction of rRNA decay.

### 4.2. Mechanistic Differences between the Current Small Molecules in Cancer Therapeutics

The effects of CX-5461 and RBI2 on cancer cells display some similarities. Only a small percentage of genes are affected by both treatments, and both result in rapid decreases in pre-rRNA levels, collectively suggesting that their effects are likely specific to ribosome biogenesis. Despite these similarities, RNA-seq analysis of the specific cohorts of genes whose expressions were altered by the two compounds suggested that the effects of RBI2 and CX-5461 on Pol I transcription and global gene expression appear to be mechanistically different.

The currently characterized ribosome biogenesis inhibitors, such as CX-5461 and BMH-21, have demonstrated the ability to effectively inhibit ribosome biogenesis through the disruption of Pol I transcription [12]. The mechanisms by which these compounds affect Pol I transcription are distinct. CX-5461 disrupts the assembly of Pol I transcription initiation complexes by interfering with the interaction of core factor SL1 with the rDNA promoter. The inhibition of Pol I transcription by CX-5461 induces anticancer effects in mice by inducing different pathways such as p53-dependent apoptosis, p53-independent senescence and autophagy, and ATM/ATR checkpoint pathways [7,9,10]. BMH-21 intercalates into GC-rich rDNA and inhibits cancer cell growth by triggering the proteasome-dependent degradation of A194 subunit, which is the largest subunit of human Pol I [12].

The findings detailed in this study suggest that RBI2 functions through a mechanism that is distinct from those of previously discovered ribosome biogenesis inhibitors. The initial evidence showing that RBI2 does not alter the Pol I occupancy of the rDNA was the first clue that RBI2 decreases cancer cell viability through a unique mechanism (Figure 3). This hypothesis was further supported by the differences elicited by RBI2 and CX-5461 in the cellular transcriptome of the MDA-MB-231 cells (Figure 4). Finally, it appears that RBI2 treatment results in an increase in polyadenylation of rRNA, which was not induced by CX-5461 (Figure 5). Though the exact mechanism of inhibition through RBI2 remains somewhat unclear, our findings indicate that RBI2 does not seem to affect the transcription initiation or elongation steps for Pol I but likely inhibits cancer proliferation by specifically upregulated rRNA turnover.

### 4.3. Potential Direct Mechanisms of RBI2-Mediated Inhibition of Cancer Cell Proliferation

In yeast it has been well established that transcription elongation via Pol I is coupled with rRNA processing [2,22,29,33]. The RBI2-treated cells contained at least 10-fold higher amounts of poly-A rRNA than the DMSO-treated control cells and the CX-5461-treated cells (Figure 5). This finding implies that such a high “background”, even after poly-dT oligo purification, is due to an increase in polyadenylated rRNA. Indeed, rRNA polyadenylation has been reported in eukaryotes such as yeast, *Candida albicans*, and mammalian cells [22,34,35]. In yeast, rRNA polyadenylation occurs in intermediate rRNA species as a consequence of impaired rRNA processing due to disrupted elongation [22]. Although mRNA is stabilized by poly(A)-tails, rRNA polyadenylation induces rRNA degradation in eukaryotes.

In yeast cells, the deletion of *RRP6*, a nuclear exosome subunit responsible for degrading aberrant RNAs, causes a dramatic accumulation of polyadenylated rRNAs [22,35]. The Trf4p/Air2p/Mtr4p complex of proteins (the TRAMP complex) acts as the nuclear surveillance complex that introduces short poly-A tails to rRNA (and other RNA), and this modification is required for the degradation of the target RNA by stimulating exosome activity [22,36].

In this study, one question remains unanswered: how does RBI2 elicit the polyadenylation of pre-rRNA? The two simplest hypotheses that may explain this observation are as follows: (1) RBI2 could induce changes in Pol I transcription, thereby eliciting the polyadenylation and degradation of pre-rRNA, or (2) RBI2 could directly activate TRAMP–exosome activity in a nucleolar-specific manner.

In support of the first model, previous studies have clearly demonstrated that alterations in the Pol I elongation rate can mediate the upregulation of pre-rRNA polyadenylation and the subsequent activation of the TRAMP–exosome pathway [22,28]. In one study, a “slow-down” point mutation in Pol I dramatically reduced the transcription elongation rate and resulted in the alternative processing and degradation of pre-rRNAs [22]. In this study, both the lack of effect of RBI2 on Pol I occupancy (Figure 3) and the lack of alternatively spliced pre-rRNAs (Figure 6) provide evidence against this model. While it is unlikely, it is certainly still possible that Pol I’s elongation rate could be altered in the RBI2-treated cells without an alteration in Pol I occupancy. For instance, if RBI2 inhibited both Pol I initiation and Pol I elongation within the same order of magnitude, one would observe no change in overall Pol I occupancy. Consistent with the model that RBI2 does not directly affect transcription elongation by Pol I, we observed no effect of the compound in fully reconstituted in vitro transcription assays (using purified proteins from *Saccharomyces cerevisiae*).

It seems more likely that the second model is true, that is, rRNA decay is directly induced without altering transcription elongation, but it is not clear how the nucleolar-specific induction of TRAMP or the exosome would be achieved. All previous TRAMP–exosome complex studies have demonstrated some manner of transcriptional stress (through changes in elongation rate, the topological stress of DNA, or through an increase in R-looping in the rDNA [22,28,30]), which subsequently signals for TRAMP recruitment. It should be noted that in these studies, alterations in Pol I occupancy on the rDNA were evident [22,28]. We did not observe direct evidence for any of these mechanisms. An additional possibility is that RBI2 treatment results in an increase in the overall abundance of the TRAMP–exosome complex. However, if this were the case, one would imagine greater changes in global transcription than those we observed in our RBI2-treatment RNA-seq experiments (Figure 4). It is also possible that RBI2 induces signaling regarding an increase in recruitment of the TRAMP complex to the nucleolus. To the best of our knowledge, this type of mechanism has not been described in any other studies. It is equally possible that an alternative RNA decay pathway is induced, exclusive of TRAMP/exosome function. For example, two of the downregulated mRNAs in all the RBI2-treated timepoints encode PABC1 and ATXN2, which are known to bind to RNA and influence its metabolism. However, the mechanism by which they could selectively influence rRNA degradation has not been revealed.

## 5. Conclusions

In the future, additional studies probing the TRAMP complex’s direct interactions with the nucleolus and pre-rRNAs in the context of ribosome biogenesis inhibition will result in a deeper understanding of this phenomenon. While the exact molecular target of RBI2 remains unclear, it is clear that RBI2 induces polyadenylation and the subsequent loss of pre-rRNA in cancer cells. This increase in rRNA polyadenylation is concurrent with a decrease in cancer cell viability, a response which was not elicited in non-transformed cells [16]. Future studies will reveal the mechanisms by which selective rRNA decay may be exploited for therapeutic benefit in cancer.

## Figures and Tables

**Figure 1 cancers-15-03303-f001:**
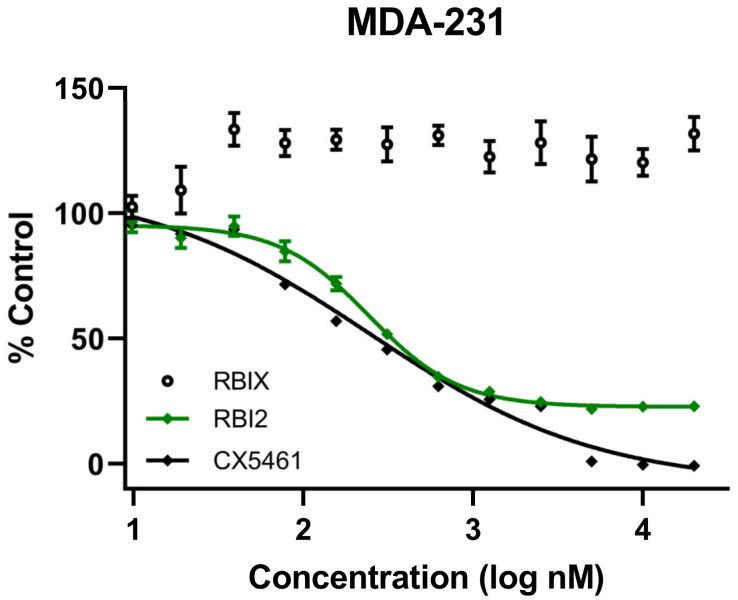
MDA-MB-231 cell viability was measured via Alamar blue assay after treatment with ribosome biogenesis inhibitor RBI2, RBIX, or CX-5461. For all treatments, cell viability at each concentration was calculated by normalizing fluorescent absorbance measurements to untreated control cells as previously described [16]. Data were plotted in GraphPad Prism. n = 3 for all treatments and error bars represent standard deviation with respect to the mean.

**Figure 2 cancers-15-03303-f002:**
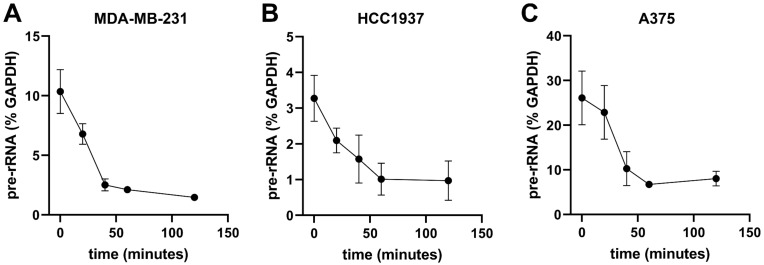
Pre-rRNA levels rapidly decreasing in cancer cell lines after treatment with RBI2. RT-qPCR was carried out in (**A**) MDA-MB-231, (**B**) HCC1937, and (**C**) A375 cancer cell lines to probe their sensitivity to treatment with RBI2. IC90 concentrations, as determined by cell growth, were used for all treatments. Lines represent linear data interpolation. n = 3 for all cell lines, and error bars represent standard deviation with respect to the mean.

**Figure 3 cancers-15-03303-f003:**
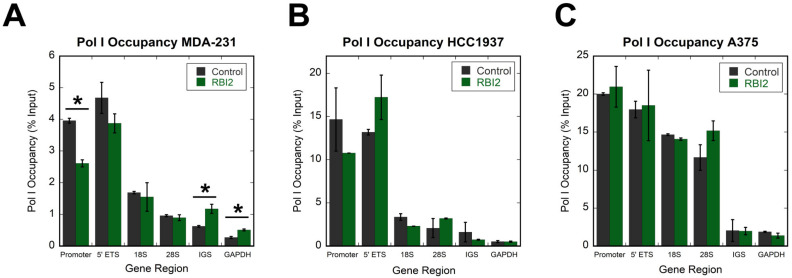
Pol I occupancy is unaltered in cancer cell lines after RBI2 treatment. (**A**) MDA-MB-231, (**B**) HCC1937, and (**C**) A375 cell lines were tested with respect to Pol I occupancy over the different regions of the rDNA after 120 min of treatment with DMSO (control, grey bars) or [IC90] RBI2 (green bars). GAPDH was used as a negative control, as Pol I does not transcribe mRNA coding genes. n = 3; error bars represent standard deviation for the mean. Statistically significant differences (*p* < 0.05) are noted with and asterisk (*). All differences between the treated and untreated samples were insignificant, with the exception of a small decrease in the promoter region in MDA-MB-231 cells.

**Figure 4 cancers-15-03303-f004:**
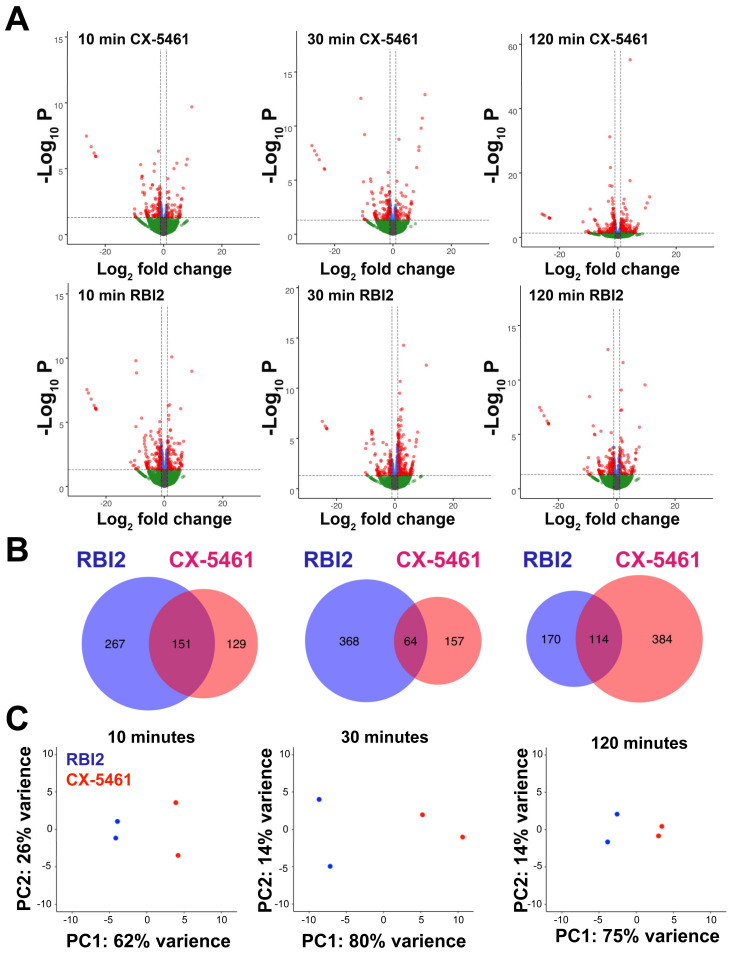
MDA-MB-231 cells respond distinctly to RBI2 and CX-5461 treatments. RNA sequencing was carried out after 10, 30, and 120 min for RBI2- and CX-5461-treated cells at a concentration of 5 µM for each compound. (**A**) Volcano plots of differential gene expression at each time point were produced for both CX-5461 (top) and RBI2 (bottom). For both CX-5461 and RBI2 treatment time-courses, n = 2. For red genes, q < 0.05 and log2 fold-change ≥1 or ≤−1; for green genes, q > 0.05, and log2 fold-change ≥1 or ≤−1. (**B**) Genes that were significantly up- or down-regulated after RBI2 or CX-5461 treatment for 10, 30, and 120 min (from left to right) were compared between treatments. (**C**) Principle component analysis of RBI2 (blue) and CX-5461 (red) treated replicates at each time point.

**Figure 5 cancers-15-03303-f005:**
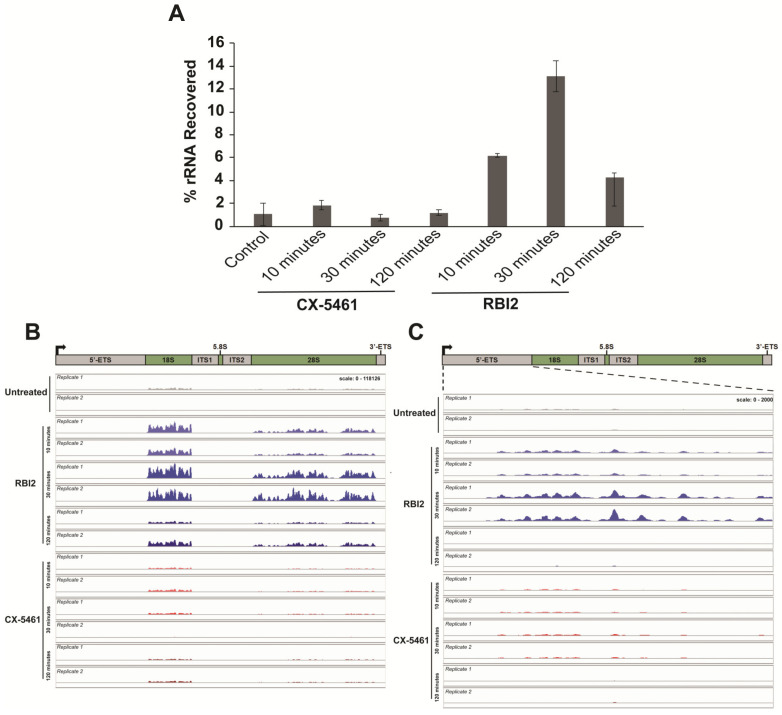
rRNA polyadenylation occurs specifically in RBI2-treated cells instead of CX-5461-treated cells and is dynamic in terms of its signal over the course of the RBI2 treatment. (**A**) rRNA concentration observed as a percentage of total RNA analyzed during RNA-seq experiments in the presence of DMSO vector (control), 5 µM of CX-5461, or 5 µM of RBI2 at 10, 30, and 120 min of treatment. (**B**) Sequence alignment of rRNA reads to the rDNA (U13369.1:1-13314). (**C**) Sequence alignment of rRNA reads to the 5′ETS region of the rDNA (U13369.1:1-3656).

**Figure 6 cancers-15-03303-f006:**
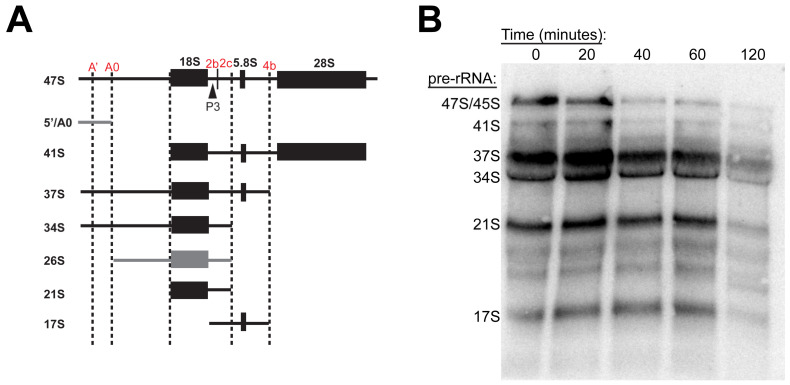
RBI2 results in an overall decrease in the mature rRNA precursors but does not appear to alter normal processing pathways for MDA-MB-231 cells. (**A**) Processing map of rRNA precursors in which P3 oligos are noted with arrows and processing sites are highlighted in red. Black pre-rRNAs signify those observed via Northern blotting in panel B, while grey symbolizes un-observed pre-rRNAs. Adapted from [23]. (**B**) Representative Northern blot probed with P3 oligo to detect rRNA precursor abundance (n = 2).

**Table 1 cancers-15-03303-t001:** Primer sequences used for RT-qPCR.

Primer Region	Sequence
rDNA 5′ ETS	Forward: 5′-GAACGGTGGTGTGTCGTT-3′ Reverse:5′-GCGTCTCGTCTCGTCTCACT-3′

**Table 2 cancers-15-03303-t002:** Primer sequences used for ChIP RT-qPCR.

Region	Sequence
rDNA Promoter	Forward: 5′-GAGGTATATCTTTCGCTCCGAGTC-3′ Reverse: 5′-CAGCAATAACCCGGCGG-3′
rDNA 5′ETS	Forward: 5′-GAACGGTGGTGTGTCGTT-3′ Reverse: 5′-GCGTCTCGTCTCGTCTCACT-3′
rDNA 18S	Forward: 5′-AAACGGCTACCACATCCAAG-3′ Reverse: 5′-CCTCCAATGGATCCTCGTTA-3′
rDNA 28S	Forward: 5′-TGGGTTTTAAGCAGGAGGTG-3′ Reverse: 5′-AACCTGTCTCACGACGGTCT-3′
rDNA IGS	Forward: 5′-TGGTGGGATTGGTCTCTCTC-3′ Reverse: 5′-CAGCCTGCGTACTGTGAAAA-3′

**Table 3 cancers-15-03303-t003:** Primers used for probing in Northern blotting.

Oligonucleotide	Sequence
P3	5′-AAGGGGTCTTTAAACCTCCGCGCCGGAACGCGCTAGGTA C-3′
P7	5′-CTTTTCCTCCGCTGACTAATATGCTTA-3′

## Data Availability

All data are available upon request. RNA seq data were deposited in GEO under the accession number GSE188262.

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
