# Peer review of "Small Molecule RBI2 Disrupts Ribosome Biogenesis through Pre-rRNA Depletion"

_cancers, 2023, doi:10.3390/cancers15133303_

Round 1

Reviewer 1 Report

Scull C. et al previously identified the small molecular drug RBI2 via high-throughput screening of inhibitors of ribosome biogenesis. In this study, they describe its application and mechanism of action. Using cell-lines, they show that RBI2 perturb cancer cell growth and viability. They also show that RBI2 does not affect Pol I occupancy of rDNA or pre-rRNA processing pathways. Rather, it increases polyadenylation of rRNAs, leading to their degradation.

Although the authors show RBI2 mediated toxicity in various cell-lines, the mechanism of action requires further detailed study and significant revision.

·      In Fig. 4 authors compare transcriptome-wide changes between RBI2 and CX-5461 drug treated cells. However, they did not check if RBI2 treated cells have consistent transcriptional changes across time-points. It would be interesting to see which genes consistently change after RBI2 treatment at different time-points.  Furthermore, if there are consistently changed genes, do they have any connection with cell proliferation, rRNAs, or polyadenylation?

·      In Fig. 5 B, C authors show poly(A) signal from rRNA locus, but it is not very convincing. Are the authors certain that the reads are indicative of poly(A)? The figure suggests that the reads overlap with 18S, 28S rRNA, and 5'ETS, rather than indicating poly(A). The authors should consider using reference data from poly-A-seq in literature to confirm the poly(A) sites for rRNAs. The poly-A-seq track from Merck in the UCSC genome browser could be helpful in this regard. Also proper annotation is necessary to know which part of the genome authors showing in the figure.

·      Do RBI2 induce polyadenylation of other RNAs (mRNAs, tRNAs, non-coding RNAs etc). Since, authors already have RNA-seq data this can be checked.

·      In Fig 6 B, the loading control is missing. It would be more confident if either a housekeeping gene was shown or a methylene blue-stained membrane was presented.

·      Authors claim that RBI2 is specific for cancer cells but this specificity is not addressed here. it is crucial to determine mechanisms of specificity of RBI2 for its future therapeutic applications. It would be interesting to investigate whether RBI2 treatment also induces polyadenylation of rRNAs in non-transformed, non-cancerous cells. 

·      Furthermore, the authors have not demonstrated if RBI2 treatment perturb protein synthesis. Conducting polysome profiling, puromycin incorporation assays, or other suitable assays would be necessary to show that RBI2 effectively reduces protein synthesis.

·      Blocking rRNA synthesis often results in P53 activation, cell-cycle arrest, autophagy. It would be valuable for the authors to investigate whether such pathways are activated by RBI2. Performing a cell-cycle assay could provide insights into the specific stage at which RBI2 induces cell-cycle arrest.

Overall, the manuscript requires attention to these points for improvement and clarification.

Author Response

We thank the editor and the reviewers for their thoughtful comments and suggestions. We have taken all comments into consideration and have revised the manuscript accordingly. Please see our detailed comments and responses (point-by-point) below.

Reviewer #1

Scull C. et al previously identified the small molecular drug RBI2 via high-throughput screening of inhibitors of ribosome biogenesis. In this study, they describe its application and mechanism of action. Using cell-lines, they show that RBI2 perturb cancer cell growth and viability. They also show that RBI2 does not affect Pol I occupancy of rDNA or pre-rRNA processing pathways. Rather, it increases polyadenylation of rRNAs, leading to their degradation.

Although the authors show RBI2 mediated toxicity in various cell-lines, the mechanism of action requires further detailed study and significant revision.

  • In Fig. 4 authors compare transcriptome-wide changes between RBI2 and CX-5461 drug treated cells. However, they did not check if RBI2 treated cells have consistent transcriptional changes across time-points. It would be interesting to see which genes consistently change after RBI2 treatment at different time-points.  Furthermore, if there are consistently changed genes, do they have any connection with cell proliferation, rRNAs, or polyadenylation?

            To address this, we have compared the set of significant differentially expressed genes across RBI2 treatment times and found 84 genes differentially expressed across all time points, all with consistent log2 fold change directions. While GO term enrichment analysis did not return any significant results, we did find two known polyA-binding proteins PABC1 and ATXN2 with negative log2 fc values. These proteins have previously been shown to interact with one another and polyadenylated mRNAs (PMID: 25721894, PMID: 15663938, PMID: 29395067). We have included a supplementary figure of the DEG gene set comparison, and a sentence noting the interesting interaction of PABC1 and ATXN2 to the results.

  • In Fig. 5 B, C authors show poly(A) signal from rRNA locus, but it is not very convincing. Are the authors certain that the reads are indicative of poly(A)? The figure suggests that the reads overlap with 18S, 28S rRNA, and 5'ETS, rather than indicating poly(A). The authors should consider using reference data from poly-A-seq in literature to confirm the poly(A) sites for rRNAs. The poly-A-seq track from Merck in the UCSC genome browser could be helpful in this regard. Also proper annotation is necessary to know which part of the genome authors showing in the figure.

            The region shown in the figure is a reference sequence for a complete human rDNA repeating unit U13369.1 as stated in methods. Reads were aligned to this reference as rDNA sequences annotations are not included in the reference genome. We are unable to use the suggested poly-A-seq reference track or other reference datasets, as they lack comparable rDNA annotations. U13369.1 coordinate ranges have been added to Figure 5 legends for clarity.

  • Do RBI2 induce polyadenylation of other RNAs (mRNAs, tRNAs, non-coding RNAs etc). Since, authors already have RNA-seq data this can be checked.

            We examined this by aligning RNA-seq reads to hg19 and viewing read coverage of poly-A-seq sites using the suggested poly-A-seq track in the UCSC genome browser. We found that sequence coverage aligned with reference poly(A) sites, but did not find evidence to suggest the enrichment of polyadenylation of other RNAs in comparison to DMSO control of CX-5461 treated samples.

  • In Fig 6 B, the loading control is missing. It would be more confident if either a housekeeping gene was shown or a methylene blue-stained membrane was presented.

To control for loading of the Northern blots, we evaluate ethidium bromides staining of the gels, which is included in the supplement. This method is more sensitive than methylene blue, in our experience. In previous experiments we see consistent 28S signal with P7 probe with analogous membranes, but destaining these radioactive membranes is technically challenging, and so it was not carried out for the specific membrane shown in figure 6B.

  • Authors claim that RBI2 is specific for cancer cells but this specificity is not addressed here. it is crucial to determine mechanisms of specificity of RBI2 for its future therapeutic applications. It would be interesting to investigate whether RBI2 treatment also induces polyadenylation of rRNAs in non-transformed, non-cancerous cells. 

In our previous work (Scull, et al. Biochemical Journal, 2019), we evaluated effects of RBI2 on non-transformed cell growth, and we also evaluated the effects of a closely related, but inactive compound (RBIX). Given that we showed no effect on growth, there is no justification to pursue further studies in the HUVEC cells.

  • Furthermore, the authors have not demonstrated if RBI2 treatment perturb protein synthesis. Conducting polysome profiling, puromycin incorporation assays, or other suitable assays would be necessary to show that RBI2 effectively reduces protein synthesis.

We agree that this could provide additional insights into the cellular response to RBI2, but these experiments are outside the scope of the current work.

  • Blocking rRNA synthesis often results in P53 activation, cell-cycle arrest, autophagy. It would be valuable for the authors to investigate whether such pathways are activated by RBI2. Performing a cell-cycle assay could provide insights into the specific stage at which RBI2 induces cell-cycle arrest.

Again, we agree that evaluating stress responses to RBI2 will be interesting. However, this manuscript focuses on the dynamic response of rRNA levels with acute treatment, thus these proposes experiments are outside the scope of the current work.

Reviewer 2 Report

This manuscript reports a unique mechanism of action by which small molecule RBI2 inhibits ribosome biogenesis. Known inhibitors of ribosome biogenesis like CX-5461 and BMH-21 work by inhibiting Pol I rDNA binding whereas RBI2 works by inducing polyadenylation of pre-rRNA (resulting in subsequent degradation of pre-rRNA). Some experiments and clarifications can improve the manuscript. Please find comments below:

Comments:

1. In these study authors observed distinct mechanism of inhibition of ribosome biosynthesis for RBI2 compared to CX-5461 and BMH-21. In light of these observations, it will be interesting to test whether these drugs (RBI2 and CX-5461; RBI2 and BMH-21) show synergy or additivity in killing cancer cells.  This can shed some light on the mechanism of action of small molecule RBI2.

2. Figure 2: Graphical representation in Figure 2C was not drawn/prepared similarly as Figure 2A or Figure 2B. Lines in Figure 2C does not look like linear interpolation as in Figure 2A or Figure 2B.

3. Figure 2A, B, C and Figure S2A, B: ‘No treatment controls’ are missing for the time points 20, 40, 60 and 120 minutes.

4. Section 3.3 (lines 208-209): Please rewrite the first line which is little confusing and misleading for the readers. There should not be any pre-rRNAs before transcription.

5. Section 3.3, Line 6: “rRNA levels” should be “pre-rRNA levels”.

6. Please include statistical tests for the bar graphs in Figures 3A, B and C.

Manuscript is well written. Minor editing required

Author Response

We thank the editor and the reviewers for their thoughtful comments and suggestions. We have taken all comments into consideration and have revised the manuscript accordingly. Please see our detailed comments and responses (point-by-point) below.

Reviewer #2

This manuscript reports a unique mechanism of action by which small molecule RBI2 inhibits ribosome biogenesis. Known inhibitors of ribosome biogenesis like CX-5461 and BMH-21 work by inhibiting Pol I rDNA binding whereas RBI2 works by inducing polyadenylation of pre-rRNA (resulting in subsequent degradation of pre-rRNA). Some experiments and clarifications can improve the manuscript. Please find comments below:

Comments:

  1. In these study authors observed distinct mechanism of inhibition of ribosome biosynthesis for RBI2 compared to CX-5461 and BMH-21. In light of these observations, it will be interesting to test whether these drugs (RBI2 and CX-5461; RBI2 and BMH-21) show synergy or additivity in killing cancer cells.  This can shed some light on the mechanism of action of small molecule RBI2.

We agree that evaluating potential synergy might be useful. However, we cannot prioritize this approach, since both BMH-21 and CX-5461 appear to rapidly inhibit transcription initiation (many papers including some from our lab). Thus, if no new rRNA is made, the effects of RBI2 would be lost.

  1. Figure 2: Graphical representation in Figure 2C was not drawn/prepared similarly as Figure 2A or Figure 2B. Lines in Figure 2C does not look like linear interpolation as in Figure 2A or Figure 2B.

Thank you for this point. We have instead represented the plots in Graphpad so that the linear interpolations are drawn consistently.

  1. Figure 2A, B, C and Figure S2A, B: ‘No treatment controls’ are missing for the time points 20, 40, 60 and 120 minutes.

Yes, all of our data are plotted normalize to time=0. In our previous work (Scull, et al. Biochemical Journal, 2019), we evaluated effects a closely related, but inactive compound (RBIX) over the time courses indicated (2 hours) and observed no effects. Due to this observation and the overall short time course of the experiment, we did not deploy untreated samples at each point. 

  1. Section 3.3 (lines 208-209): Please rewrite the first line which is little confusing and misleading for the readers. There should not be any pre-rRNAs before transcription.

We thank the reviewer for suggesting a change in language here, we agree - it was confusing. We have amended this sentence.

  1. Section 3.3, Line 6: “rRNA levels” should be “pre-rRNA levels”.

This has been corrected.

  1. Please include statistical tests for the bar graphs in Figures 3A, B and C.

We have included statistics as suggested. While we do see for one cell line (MDA-231) that the data appear significant for a difference in Pol I occupancy at the promoter, this is likely due to the rather low signal at this position rather than a true biological phenomenon – this cell line showed the smallest amount of Pol I occupancy in either the control or the treated samples.

Round 2

Reviewer 1 Report

The authors have not adequately addressed some of the comments. 

  • In Fig. 5 B, C authors show poly(A) signal from rRNA locus, but it is not very convincing. Are the authors certain that the reads are indicative of poly(A)? The figure suggests that the reads overlap with 18S, 28S rRNA, and 5'ETS, rather than indicating poly(A). The authors should consider using reference data from poly-A-seq in literature to confirm the poly(A) sites for rRNAs. The poly-A-seq track from Merck in the UCSC genome browser could be helpful in this regard. Also proper annotation is necessary to know which part of the genome authors showing in the figure.

            The region shown in the figure is a reference sequence for a complete human rDNA repeating unit U13369.1 as stated in methods. Reads were aligned to this reference as rDNA sequences annotations are not included in the reference genome. We are unable to use the suggested poly-A-seq reference track or other reference datasets, as they lack comparable rDNA annotations. U13369.1 coordinate ranges have been added to Figure 5 legends for clarity.

     Authors response does not fully answer the raised query regarding the identification of reads as poly(A) instead of 18S, 28S rRNA, and ETS. Authors should make sure that reads are really are of poly(A) and not of other part of the rRNA as shown in the fig 5B,C. It is important to include a separate figure displaying nucleotide resolution, specifically highlighting a Poly (A) portion of the locus with nucleotide labels. Authors can repeat the analysis with other reference rDNA locus which has annotated poly (A) sites in Merck data (or any other positive control data). Or authors should find a better strategy to convince that the reads are of poly (A) and also for proper representation.

·       Do RBI2 induce polyadenylation of other RNAs (mRNAs, tRNAs, non-coding RNAs etc). Since, authors already have RNA-seq data this can be checked.

We examined this by aligning RNA-seq reads to hg19 and viewing read coverage of poly-A-seq sites using the suggested poly-A-seq track in the UCSC genome browser. We found that sequence coverage aligned with reference poly(A) sites, but did not find evidence to suggest the enrichment of polyadenylation of other RNAs in comparison to DMSO control of CX-5461 treated samples.

Which other RNAs checked for poly (A)? How many RNAs checked? Authors should ideally check this for entire transcriptome. This should be added as a figure in the manuscript with proper description.

Author Response

Please see attached response, thank you!

Round 3

Reviewer 1 Report

Authors comments are satisfactory. Manuscript can be accepted without addition of new figures presented during the 2nd round of revision.